# Self-collected and clinician-collected anal swabs show modest agreement for HPV genotyping

Racheal S. Dube Mandishora[1,2]*, Trine B. Rounge[3,4], Megan Fitzpatrick[5], Irene Kraus Christiansen[6,7], Ole Herman Ambur[8], Sonja Lagström[3,6], Babill Stray-Pedersen[9†], Massimo Tommasino[2], Joel Palefsky[10‡], Zvavahera M. Chirenje[11‡]

1 Department of Medical Microbiology, University of Zimbabwe College of Health Sciences, Harare, Zimbabwe, 2 Infections and Cancer Biology Group, International Agency for Research on Cancer, Lyon, France, 3 Department of Research, Cancer Registry of Norway, Oslo, Norway, 4 Department of Informatics, University of Oslo, Oslo, Norway, 5 Department of Pathology, University of Wisconsin, Madison, Wisconsin, United States of America, 6 Department of Microbiology and Infection Control, The Norwegian HPV Reference Laboratory, Akershus University Hospital, Lørenskog, Norway, 7 Department of Clinical Molecular Biology (EpiGen), Division of Medicine, Akershus University Hospital and University of Oslo, Lørenskog, Norway, 8 Faculty of Health Sciences, OsloMet—Oslo Metropolitan University, Oslo, Norway, 9 Women's Clinic, Rikshospitalet, Oslo University Hospital and Institute of Clinical Medicine, Oslo, Norway, 10 Department of Medicine, UCSF School of Medicine, San Francisco, CA, United States of America, 11 Department of Obstetrics and Gynaecology, University of Zimbabwe College of Health Sciences, Harare, Zimbabwe

† Deceased.
‡ These authors share senior authorship on this work.
* racheal.mand@gmail.com

**Data Availability Statement:** All relevant data are within the manuscript and its Supporting Information files.

## Abstract

### Background & aim

Women with HIV/HPV coinfection and cervical lesions are at increased risk of developing HPV related anal cancer. Self-collection of anal swabs may facilitate HPV molecular testing in anal cancer screening, especially in high-risk groups, and yet it is not adequately studied. We evaluated level of agreement between self-collected anal swabs (SCAS) and clinician-collected anal swabs (CCAS) when used for HPV genotyping. We also described the anal HPV genotype distribution and HIV/HPV coinfection.

### Methods

We performed a cross sectional study with participants from a visual-inspection-with-acetic-acid and cervicography (VIAC) clinic, in Harare, Zimbabwe. In a clinic setting, the women aged ≥18 years provided anal swabs in duplicate; first CCAS and then SCAS immediately after. HPV detection and genotyping were performed using next generation amplicon sequencing of a 450bp region of the HPV L1 gene. Level of agreement of HPV genotypes between CCAS and SCAS was calculated using the kappa statistic. McNemar tests were used to evaluate agreement in the proportion of genotypes detected by either method.

**Funding:** Two organisations funded the research reported in this publication; Letten Foundation Norway and The Fogarty International Center of the National Institutes of Health under Award Number D43TW010137 within the University of Zimbabwe PERFECT program.

**Competing interests:** The authors have declared that no competing interests exist.

## Results

Three-hundred women provided 600 samples for HPV genotyping. HPV genotypes were detected in 25% of SCAS and in 22% of CCAS. The most common genotypes with CCAS were HPV52, HPV62 and HPV70 and with SCAS were HPV62, HPV44, HPV52, HPV53 and HPV68. Total HPV genotypes detected in CCAS were more than those detected in SCAS, 32 versus 27. The agreement of HPV genotypes between the two methods was 0.55 in kappa value (k). The test of proportions using McNemar gave a Chi-square value of 0.75 (p = 0.39). Multiple HPV infections were detected in 28/75 and 29/67 women for CCAS and SCAS respectively.

## Conclusions

SCAS and CCAS anal swabs showed moderate agreement, with no statistically significant difference in the proportion of genotypes detected by either methods. Although the differences between the two methods were not statistically significant, CCAS detected more HPV genotypes than SCAS and more HPV infections were detected in SCAS than in CCAS. Our data suggest that self-collected anal swabs can be used as an alternative to clinician-collected anal swabs for HPV genotyping.

## 1. Introduction

Human papillomaviruses (HPVs) are the most common viral infections in the genital tract and are the main etiologic agent of anogenital cancers, such as cervical, anal, vulval, penile and vaginal malignancies [1]. Although development of cancer only occurs in a small percentage of individuals who are infected with HPV, the high prevalence of the virus and its severe consequences makes HPV-related cancers a high-priority research area [2]. HPV accounts for almost all new cervical and anal cancer cases, as reported in 2018, with most of the cases disproportionately occurring in low- and middle-income countries. Zimbabwe reports an age-standardised cervical cancer incidence rate of 62.3/100000, ranking it as the 4th highest globally [3]. HPV-related cancers have been widely studied. Women who acquire cervical HPV infections may shed the virus at anatomical sites of vagina, vulva and anus, thus one anogenital site may act as a reservoir for HPV infection in other sites [4]. Women with a history of cervical cancer, especially those with HPV/HIV co-infection, are at increased risk of developing anal cancers [5]. Screening for anal cancer is thus recommended for women with cervical lesions and/or who are HIV infected [6, 7].

Although anal cancers are relatively rare, data from USA indicates a significant increase in the incidence of anal cancers [8]. The increase is particularly high among HIV-infected individuals despite the availability and widespread use of anti-retroviral-therapy (ART), which improves individuals' immunogenic profile. Conversely, ART extends life among HIV infected individuals, giving time for cancer to develop [9]. Zimbabwe has an HIV prevalence of almost 13% [10] and cervical cancer contributes a third of all cancers among black Zimbabwean women [11]. In 2016, the Zimbabwe Cancer Registry reported a gradual increase in anal cancers recorded in the past decade. Despite the increased prevalence of risk factors in the population and evidence of increasing incidence, anal cancer screening guidelines are not routinely followed in Zimbabwe.

Utility of self-sample cervical swabs for HPV genotyping has been demonstrated [12]. Furthermore, self-collection of cervical swabs has shown to be highly acceptable [13, 14]. For anal screening, there is evidence that self-sampling is acceptable and patients tend to be more comfortable and less embarrassed when an anal swab is self-collected [15]. Its clinical utility has been demonstrated for non-HPV studies [16, 17] and for anal cytology [18–20]. However, there is still scarce data with regards to self-collected sample for HPV DNA testing. Improvements in the performance of molecular based tests permit consideration of self-collection as a promising approach to increase anal HPV screening [21].

Prioritising research on HPV-related anal cancers in women cannot be over emphasised. Most published data on HPV and anal malignancies is on men-who-have-sex-with men (MSM) [18, 22–24]. Heterosexual women have been neglected and they are more vulnerable to infection with oncogenic HPV. This is because of their anatomy, which allows the inoculation of HPV from the vaginal region to the anus even without practicing anal intercourse [25]. Furthermore, understanding the different collection methods will help researchers and clinicians to prepare for future application of anal cancer screening.

Against this background, our study primarily aimed at determining the level of agreement of self-collected anal swabs (SCAS) and clinician-collected anal swabs (CCAS) for anal HPV genotyping, in women reporting for routine cervical cancer screening in a clinical setting. We also reported the anal HPV genotype distribution and HIV co-infection in this group of women who are at increased risk of anogenital cancers, and as additional epidemiologic data. The description of HPV genotype distribution is important for clinical recommendations, for surveillance before and after introducing HPV vaccination and for comparison of the HPV in circulation in both the cervix and anus [26].

## 2. Methods

### 2.1 Study design and population

The study was cross-sectional and recruited women reporting for routine cervical cancer screening from the general population. The cervical cancer screening was at a Visual Inspection with Acetic-acid and cervicography (VIAC) clinic within a tertiary hospital, Parirenyatwa Hospital, situated in the capital city of Zimbabwe, with the sample set representing a populations in the city and in the rural and urban peripheries of the city. The questionnaire was available in both English and Shona (S1 File).

The estimated sample size was 300 women, accounting for an assumed 7% attrition rate. The samples were collected for a primary objective to obtain HPV genotype specific population parameters, such as prevalence and genetic evolution of the virus. The A. J Dobson's formula for prevalence was used, where the estimated prevalence was 24% based on known prevalence of HPV genotype detection from a previous study done in Zimbabwe [27]. The confidence interval was set at 95% with a desired width of 5%, giving a total of 280 women. Twenty women were included to cover any attrition. All the samples were then used to address the secondary objective of comparing self-collected and clinician-collected anal swabs for HPV genotyping.

Ethics approval was obtained from The Joint Parirenyatwa hospital and College of Health Sciences Research Ethics Committee (reference: JREC210/14), Medical Research Council of Zimbabwe (reference: MRCZ/A/1911) and Research Council of Zimbabwe RCZ (Permit number 03064).

### 2.2 Recruitment and laboratory methods

Written informed consent, in English or Shona, was obtained from the women who were ≥18 years, sexually active and had no history of a total abdominal hysterectomy. Before consenting,

they were taken through the study protocol and were made aware of future publications, which would be written with non-identifiers or with no information that could compromise the participant's identity.

Enrolment was from February to April 2015 and the recruitment process is previously described [26]. Anal Dacron[TM] swabs [28, 29] were collected and VIAC, Pap-smear and HIV testing were performed on these women, as previously described [30]. Briefly, sexually active women aged at least 18 years old, reporting for routine cervical cancer screening at the VIAC clinic were consecutively recruited. Women who did not consent to providing duplicate anal swabs were excluded. Two anal swabs were requested from each woman; one SCAS and one CCAS. The nurse collected the CCAS in the examination room, by gently inserting the swab into the anal canal until the shaft could not move further and rotated it for 10–30 seconds. The nurse explained the self-collection procedure to the women, who then immediately proceeded to the toilet within the clinic facility for the SCAS collection. All swabs had Dacron[TM] tips and a firm plastic shaft. The swabs were immediately broken into a cryotube soon after collection; SCAS were broken by the participant and the CCAS by the collecting clinician. All swabs were stored in 500μl lysis buffer from bioMerieux (containing guanadine thiocyanate) at -80˚C until analysed.

DNA was extracted from the anal swabs using the ZR Genomic DNA[TM] Tissue MiniPrep kit, ZYMO Research, USA, as per manufacturer's instructions [31]. The eluted DNA was stored at -20˚C, awaiting next-generation-sequencing. For quality control, betaglobin testing was performed to verify DNA quality using conventional PCR in Zimbabwe before samples were sent for next-generation sequencing in Norway.

**2.2.1 Next-generation-sequencing (NGS).** The detection and genotyping of HPV was done by amplification of a 450 bp amplicon from the HPV L1 region, using the Illumina MiSeq platform, as previously described [26]. Briefly, using PGMY primers (Table 1) [32], amplicons with Illumina-tail were generated in 20 μl volumes using Phusion Master Mix

**Table 1. List of PGMY primers used for Illumina next-generation-sequencing.**

| Primer name | Locus specific sequence | Sequence with Illumina adapter |
|---|---|---|
| PGMY11-A | GCACAGGGACATAACAATGG | ACACTCTTTCCCTACACGACGCTCTTCCGATCTGCACAGGGACATAACAATGG |
| PGMY11-B | GCGCAGGGCCACAATAATGG | ACACTCTTTCCCTACACGACGCTCTTCCGATCTGCGCAGGGCCACAATAATGG |
| PGMY11-C | GCACAGGGACATAATAATGG | ACACTCTTTCCCTACACGACGCTCTTCCGATCTGCACAGGGACATAATAATGG |
| PGMY11-D | GCCCAGGGCCACAACAATGG | ACACTCTTTCCCTACACGACGCTCTTCCGATCTGCCCAGGGCCACAACAATGG |
| PGMY11-E | GCTCAGGGTTTAAACAATGG | ACACTCTTTCCCTACACGACGCTCTTCCGATCTGCTCAGGGTTTAAACAATGG |
| PGMY09-F | CGTCCCAAAGGAAACTGATC | AGACGTGTGCTCTTCCGATCTCGTCCCAAAGGAAACTGATC |
| PGMY09-G | CGACCTAAAGGAAACTGATC | AGACGTGTGCTCTTCCGATCTCGACCTAAAGGAAACTGATC |
| PGMY09-H | CGTCCAAAAGGAAACTGATC | AGACGTGTGCTCTTCCGATCTCGTCCAAAAGGAAACTGATC |
| PGMY09-I | GCCAAGGGGAAACTGATC | AGACGTGTGCTCTTCCGATCTGCCAAGGGGAAACTGATC |
| PGMY09-J | CGTCCCAAAGGATACTGATC | AGACGTGTGCTCTTCCGATCTCGTCCCAAAGGATACTGATC |
| PGMY09-K | CGTCCAAGGGGATACTGATC | AGACGTGTGCTCTTCCGATCTCGTCCAAGGGGATACTGATC |
| PGMY09-L | CGACCTAAAGGGAATTGATC | AGACGTGTGCTCTTCCGATCTCGACCTAAAGGGAATTGATC |
| PGMY09-M | CGACCTAGTGGAAATTGATC | AGACGTGTGCTCTTCCGATCTCGACCTAGTGGAAATTGATC |
| PGMY09-N | CGACCAAGGGGATATTGATC | AGACGTGTGCTCTTCCGATCTCGACCAAGGGGATATTGATC |
| PGMY09-P | GCCCAACGGAAACTGATC | AGACGTGTGCTCTTCCGATCTGCCCAACGGAAACTGATC |
| PGMY09-Q | CGACCCAAGGGAAACTGGTC | AGACGTGTGCTCTTCCGATCTCGACCCAAGGGAAACTGGTC |
| PGMY09-R | CGTCCTAAAGGAAACTGGTC | AGACGTGTGCTCTTCCGATCTCGTCCTAAAGGAAACTGGTC |
| RSMY09-L | CGTCCTAATGGGAATTGGTC | AGACGTGTGCTCTTCCGATCTCGTCCTAATGGGAATTGGTC |
| HMB01 | GCGACCCAATGCAAATTGGT | AGACGTGTGCTCTTCCGATCTGCGACCCAATGCAAATTGGT |

(Thermo Fischer Scientific, MA, USA), 0.1 μM of each primer and 5 μl sample under the following conditions; 98˚C for 30 seconds, 40 cycles of 98˚C for 10 seconds, 56˚C for 30 seconds and 72˚C for 15 seconds, before a final extension 72˚C for 10 minutes. Amplicons were then used as templates in indexing PCR in 20 μl volumes using Phusion Master Mix, 0.375 μM each index [33] and 1 μl template under the following conditions; 98˚C for 30 seconds, 8 cycles of 98˚C for 10 seconds, 65˚C for 30 seconds and 72˚C for 20 seconds before a final extension 72˚C for 5 min. The resulting amplicon libraries were pooled together in equal volumes, cleaned up twice using 0.7 × AMPure XP (Agencourt Beckman Coulter, CA, USA).Library quality control and quantitation was performed on the Agilent 2100 Bioanalyzer using Agilent High Sensitivity DNA Kit (Agilent Technologies, CA, USA) and by qPCR using KAPA DNA library quantification kit (Kapa Biosystems, Wilmington, MA). For HPV detection and genotyping, the libraries containing the 450 bp PGMY amplicons from the HPV L1 region were sequenced on the MiSeq platform (Illumina, CA, USA) using V2 chemistry and 2 × 250 bp reads.

**2.2.2 Bioinformatics and statistics.** We counted HPV sequences per genotype for each sample using a custom data analysis pipeline. Adapters, primers, short reads (<50 bases) and low-quality bases (<15) removed with Cutadapt (version 1.8.3). The sequences were mapped to an HPV genome database using Bowtie2 v 2.2.9 in an "end-to-end" mode and pre-set "sensitive" setting and only the best alignments were reported [30]. The HPV genome database consisted of 183 reference HPV genomes obtained from PaVe [34]. Number of read pairs mapped to each genome were counted by FeatureCount (version 1.4.6) from the Subread package. Further data cleaning and statistical analyses were performed on R (v3.6.0). A final cut-off of 500 mapped read pairs or more were considered as positive [35–37]. Choice of cut-off was based on the mean reads of CCAS and SCAS. Frequencies were calculated as a percentage of the total. Plots of HPV genotype distribution and agreement of the two methods were made using frequencies of each HPV genotype. Sequence read counts for genotypes detected as one of the most frequent, on both CCAS and SCAS, were tabulated for comparison. This was to check if the amount of HPV collected by the first swab differed from that collected by the second swab in the same individual.

Using R, data were subset into CCAS and SCAS. Categorical variables were created, of either 'positive' for all samples with a read count >499 or 'negative' for those with read counts <499. Bar plots were drawn to illustrate frequencies of individual HPV genotypes. To evaluate agreement based on proportions the two collection methods, the McNemar non-parametric test was performed. A kappa test was used to evaluate agreement of HPV genotypes across the two collection methods. The McNemar's non-parametric test was performed to evaluated agreement in the proportion of genotypes detected by either method [38–40]. Agreement was either when both SCAS and CCAS, from the same woman, were positive for a specific HPV genotype or when both SCAS and CCAS were negative for any HPV genotype.

## 3. Results

### 3.1 Summary demographics

A total of 300 women were enrolled into the study. These women provided 600 samples for HPV genotyping. Half (50.3%) of the women were HIV infected (Table 2). The age range of these women was 18–83 years old, with a mean of 39.7 years, with no significant differences among the HIV infected and HIV un-infected women (p = 0.13). VIAC positivity was detected in 10% of the women with more HIV-infected women testing positive at a higher rate as compared to the HIV-uninfected (p = 0.036). Similarly, of the 217 women who had normal cytology outcomes, over half (122/217) were HIV-uninfected.

**Table 2. Demographic and surveillance data of participants, grouped by their HIV status.**

| Category | All participants n = 300 *a/n is total detected over total women with available data | HIV Positive n = 151 n(%)except* | HIV Negative n = 149 n (%) *except | P-value |
|---|---|---|---|---|
| Age in years Mean (SD) | *39.7 (10.7) | *38.7 (8.1) | *40.6 (12.8) | 0.13 |
| Parity, number of children Median (IQR) | *3.0 (0–10) | *3.0 (2–4) | *3.0 (2–4) | 0.12 |
| Sexual debut in years Median (IQR) | *19 (18–22) | *19(17–21.5) | *20 (18–22) | 0.03 |
| Use of contraception a/n (%) Yes Oral tablets | 168/298 (56%) 79/168 (46.7) | 84/151 (55.6%) 29/84 (34.5) | 84/147 (57.1%) 50/84 (58.8) | 0.79 |
| History of STI a/n (%) | 98/298 (32.7%) | 62/151 (41.1%) | 36/147 (24.2%) | <0.01 |
| Multiple sexual partners at recruitment a/n (%) Participant Partner | 25/299 (8.4) 38/240 (15.8) | 16/151 (10.6) 21/115 | 9/148 (6.1) 17/125 (1) | 0.16 0.31 |
| Partner circumcised a/n (%) | 52/227 (22.9) | 24/106 (22.6) | 28/121 (23.1) | 0.93 |
| VIAC positive a/n (%) | 30/299 (10.0) | 21/151 (13.9) | 9/148 (6.0) | 0.0361 |
| Cytology normal a/n (%) | 217/283 (76.6) | *95/145 (65.5) | *122/137 (89.1) | <0.0001 |

This table gives a summary of demographics and survey data of 300 women, compiled from a structured questionnaire completed by a research nurse who interviewed each participant. VIAC = visual inspection with acetic acid and cervicography. IQR = Interquartile range. SD = Standard Deviation. STI = Sexually transmitted infection.

## 3.2 Anal HPV detection and genotyping

All samples were positive for betaglobin. SCAS generated 5 694 724 sequence read pairs whilst CCAS generated 7 843 298 (Table 3). HPV genotypes distribution showed that HPV52 were most frequent in CCAS and HPV62 most frequent in SCAS.

HPV genotypes were detected in SCAS from 75/300 women (25%) and in CCAS from 67/300 women (22%). The most common genotypes in women with positive SCAS were HPV 62 (15%) and 11% for each of HPV 44, HPV 52, HPV 53 and HPV 68, whilst in women with positive CCAS HPV 52 (13%), HPV 62 (12%) and HPV 70 (10%) were the most prevalent types (Fig 1). The agreement of HPV genotypes between the two methods was 0.55 in kappa value (k). The test of proportions using McNemar gave a Chi-square value of 0.75 (p = 0.39), when comparing all the HPV genotypes detected. The total number of HPV genotypes detected in CCAS was more than those detected in SCAS, 32 versus 27 (Fig 1). HPV 31, HPV 42, HPV 54, HPV 84, HPV 86 and HPV 87 were only detected in CCAS. Genotypes detected in both CCAS and SCAS in perfectly equal frequencies are HPV 26, HPV 30 and HPV 32 (Fig 1).

Multiple HPV infections were detected amongst 29/67 (43%) and 28/75 (37%) of CCAS and SCAS respectively.

**Table 3. Summary of sequence read pairs generated and number of positive clinician-collected (CCAS) and self-collected (SCAS) specimens.**

| Collection method of anal swab | Total read pairs (n = 300) | Mean read pairs | Max read pairs | Min read pairs | Number of HPV positive women |
|---|---|---|---|---|---|
| Clinician-collected (CCAS) | 7 843 298 | 26 232 | 104 980 | 144 | 67 |
| Self-collected (SCAS) | 5 694 724 | 19 110 | 73 905 | 50 | 75 |

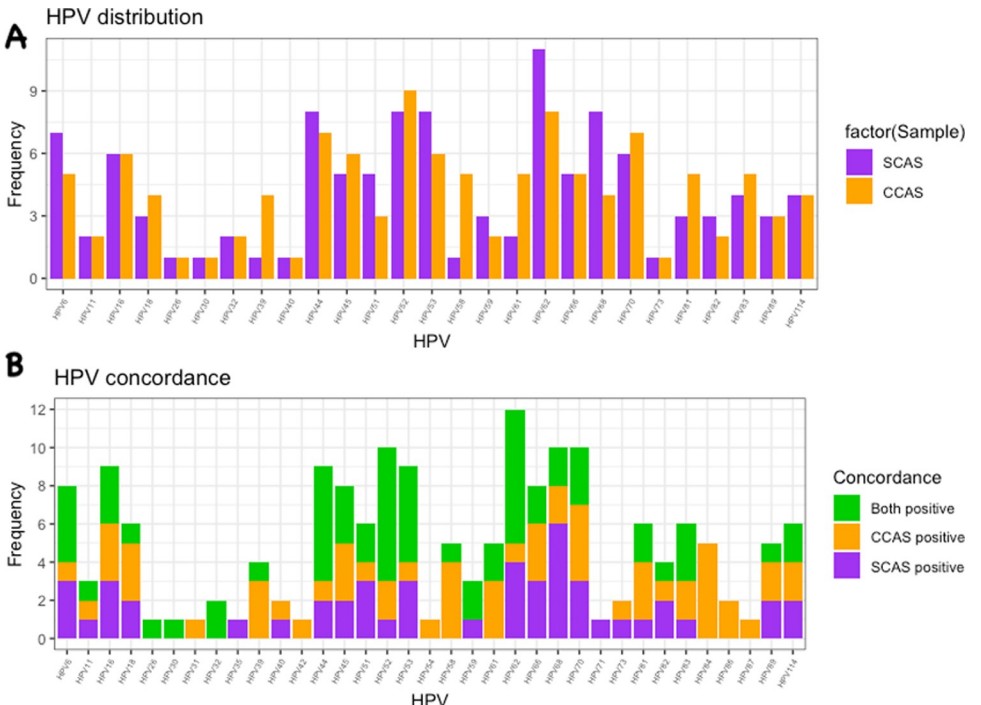

**Fig 1.** A) HPV genotypes detected in self-collected (SCAS) and clinician-collected (CCAS) anal swabs from women reporting for routine cervical cancer screening. B) Genotype concordance between SCAS and CCAS.

Sequence read counts for the genotypes that were amongst the most frequently detected, HPV 52 and HPV 62, were compared (S1 Table). A total of 12/20 (60%) CCAS samples and/or SCAS showed lower SCAS than CCAS reads.

### 3.3 HIV stratification: Anal HPV genotypes detected in clinician-collected swabs

A total of 151 (50.3%) women were HIV-infected, of which 47 (31%) were positive on HPV genotyping. A total of 32 HPV genotypes were detected in this group of women. The most common genotypes were HPV 52 (17%), HPV 16 (13%) and HPV 62 (13%), with 24 (51%) women having multiple HPV infections. A total of 149 women were HIV-uninfected and 7 (5%) were positive for any HPV genotype. Eight different HPV genotypes were detected in these HIV-uninfected women (HPV 6, HPV 39, HPV 45, HPV 52, HPV 58, HPV 66, HPV 81 and HPV 89) with HPV 66 detected in two women. Only one HIV-uninfected woman had multiple HPV infections, with three HPV genotypes (HPV 58, HPV 66 and HPV 89).

## 4. Discussion

To the best of our knowledge this is one of the few studies globally and the first from Zimbabwe to compare the level of agreement of SCAS and CCAS anal swabs for HPV genotyping. We also describe the distribution of CCAS HPV genotypes, stratified by HIV status, in the anal canal of women reporting for routine cervical cancer screening in a clinical setting in Harare, Zimbabwe. The value of our study is to provide evidence on alternative collection methods for anal samples that can be used for anal cancer screening in high-risk groups, such as HPV/HIV co-infected women and/or with a history of cervical lesions. No comparable

literature could be retrieved for anal swab collection methods, specifically on SCAS and CCAS for HPV genotyping, as most studies have focused on acceptability of anorectal swabs for other STI tests [16, 17, 41]. Very few studies compared anal self- and clinician-collection for cytology and not specifically for HPV genotyping [18, 19]. Our data addresses this critical gap of knowledge.

SCAS and CCAS show moderate agreement between the specific HPV genotype detected in these duplicate samples. The difference in detection rates of any HPV genotype in SCAS versus CCAS (25% vs 22%) could be attributed to the complex topology of the anal canal; with such complex anatomy, reproducibility may not be easily achieved, even in a scenario where clinician-collected duplicate samples are compared. Based on the moderate level of agreement of HPV genotypes between the two methods, indicated by 0.55 in kappa value (k) and a non-significant test of proportions, our data suggest that researchers can use either methods for collection. It is important for researchers to note one limitation, the use of the Kappa test alone does not warrant clinical recommendations for SCAS to be used as an alternative to the CCAS, because a kappa value (k) of at least 0.80 is usually required to provide strong support [38]. We however recommend a bigger study that also evaluates fewer, clinically relevant HPV genotypes to strengthen our inferences. For the purposes of our current aim, the analyses of all HPV genotypes were useful although it gives a limited picture for those seeking to visualise agreement of only the clinically relevant genotypes. As is, our levels of agreement are only moderate and not excellent, but are sufficient enough to give perspective of instances where NGS is used for HPV genotyping especially in research.

Self-collection is a good alternative because it is the most practical and less resource demanding method, which participants are less reluctant to perform. Although our current study was performed in a clinical setting, we also recommend future study designs that ensure that patients also collect samples from home and the use of routine laboratory genotyping tools. NGS is a highly sensitive method but is not widely used for routine diagnosis.

Distribution of HPV genotypes in the anal canal in the group of women studied here was diverse; with a total of 32 different HPV genotypes detected. Low-risk (LR) HPV genotypes were the most frequently detected, using both collection methods. HPV 52 was the only oncogenic high-risk (HR) HPV genotype among those commonly detected, regardless of HIV status. HPV 16 was noted as a common genotype only in the HIV-infected subgroup. In a previous study on a smaller number from the same group of women [26], HPV 52 was detected as one of the most common genotypes on both the cervico-vaginal and anal canal. LR-HPV genotypes such as HPV 6 and HPV 11 have been widely detected in individuals with anal or genital warts [42, 43]. The presence of HPV 53, a probable high-risk type [44], as one of the frequently detected anal HPV genotypes, agrees with previous findings from another Zimbabwean study that described HPV genotypes in the anus of men and women [43].

The total number HPV genotypes detected in CCAS was more than those detected in SCAS, 32 versus 27 (Fig 1). Furthermore, the most common HPV genotypes detected on CCAS and SCAS were slightly different. This may have occurred because the CCAS swabs were collected before the SCAS. We have several hypotheses to this effect, which are speculative and should be further investigated in future studies. The first one is that, given the decrease in read counts for over half of the HPV 52 and HPV 62 samples (S1 Table), the differences in collection methods maybe due to the fact that CCAS collects most of the exfoliated cells thus reducing the HPV yield for the SCAS that only collects the remaining bits. In addition, the SCAS may be touching different parts of the anal canal where there are different HPV genotypes from the parts touched by the CCAS. We recommend a different design for future studies, to address the possibility of HPV yield decreasing after the first swab has been collected. An ideal design would be to alternate the order of collection, for a proportion of SCAS and CCAS.

Secondly, in terms of numbers of samples with HPV positivity, SCAS detected more HPV than CCAS. We postulate that on collection of the first swabs, HPV from the surface epithelium may have been harvested, exposing more basal cells with HPV, which were easily harvested by the second swab, SCAS. The likelihood of this theory is reduced by the fact that swab collection is not invasive enough to cause aggressive abrasions on the surface epithelium. Our third, but most unlikely postulate is that contamination occurred during sample collection, accounting for the differences observed. Participants might have touched the cervico-vaginal area with the tip of the anal self-collection swab. Arbyn et al had similar observations, reporting HPV detection rates 2.28 times higher in self-collected versus clinician-collected vaginal swabs [45].

On average, 40% of the HPV infected women had multiple HPV infections, using either sample collection methods. This is a relatively high proportion of multiple HPV infections and is mostly observed for women with HPV/HIV co-infection, similar to studies from South-Africa and Zambia [43, 46]. The prevalence of HIV in the sample population was relatively high (50.3%). This is mainly attributed to the fact that a huge proportion of the women reporting for routine cervical cancer screening, at this tertiary hospital, would be coming from the HIV opportunistic clinic at the same premises. HIV-infected women are likely more informed of their increased risk of developing cervical cancer, thus they follow the screening guidelines prescribed for them.

In the context of the secondary data on the background of study participants, gathered on HIV status and cervical cancer screening, that gives a background of the study participants, HPV was more prevalent in HIV-infected women than in the HIV-un-infected group (31% vs 5%). This finding aligns with literature, describing how immunosuppression promotes persistent HPV infection and poorer clearance of the virus [47–50]. Important to note is that the oncogenic type, HPV 16, was only detected in the HIV-infected women and HPV 52 was detected in both groups. Given the small number of HIV-uninfected women with HPV, the proportion of HPV 52 cannot be elucidated to significance. On the other hand, HPV 52 and HPV 16 each make-up proportions of 17% and 13% of all the genotypes detected, indicating an increased risk of developing cancer for these women.

In addition to this, about half of these HPV/HIV co-infected women 21/47 (Table 2) also had positive VIAC results, signifying possible presence of pre-cancer or cancer lesions. Furthermore, the proportion of normal cervical cytology was higher in HIV-uninfected than the HIV-infected (Table 2). To further highlight risk of disease development, we report multiple HPV infections in half of the co-infected women versus only one woman amongst the HIV-uninfected. The HIV-infected group also exhibited significantly higher risk factors (Table 2), such as sexual debut (p = 0.03) and history of STIs (p <0.01).

To highlight additional limitations of our study, we have used both the Kappa and McNemar tests which are commonly used together to complement each other. Indeed, as mentioned by Ranganathan et al, a non-significant McNemar test alone is not sufficient to indicate good agreement. However, it is still a useful statistic, if interpreted correctly [51]. In our case, the McNemar test indicated that the proportion of discordant pairs (ie +/- vs -/+) was not statistically significant, providing useful information for evaluating (dis)agreement of HPV genotype detection.

We also could not properly measure whether HPV yield decreased when CCAS was collected before SCAS. Therefore, we recommend future studies that alternate the order of collection. Another possible limitation is that a few of the samples may have been reported as false negatives. Due to limitation of funds, a different test could not be used to verify this, but considering the mean and maximum reads recorded for both CCAS and SCAS (Table 3), and the positive betaglobin PCR test that verified DNA quality we are confident that a negligible number of women may have fallen within the false negative range.

## 5. Conclusion

SCAS and CCAS anal swabs showed moderate HPV genotype agreement, with non-statistically significant difference. SCAS also detected more HPV infections than CCAS, although CCAS detected a higher number of different HPV genotypes than SCAS. Our data suggest that self-collection of anal swabs can be used as an alternative to clinician collection for HPV genotyping in a clinical setting, especially where NGS is being used for research purposes. The diverse anal HPV genotype distribution detected, mostly in the HIV-infected who also had positive VIAC results, supports the recommendation for introducing routine anal cancer screening among this high-risk group.

## Supporting information

**S1 File.**
(PDF)

**S1 Table. Comparison of sequence read counts for HPV 52 and HPV 62 in both CCAS and SCAS.**
(DOCX)

## Acknowledgments

We would like to acknowledge Maria Da Costa (UCSF), who consistently gave counsel during sample collection and DNA extraction, Vasco Chikwasha for his statistical expertise, which he contributed during the proposal writing stages and Eugénie Lohmann for her contribution with R-studio commands. We also express our gratitude to Sister Mucheche, the nurse who recruited the women, the personnel at UZCHS-CTRC laboratory, the Norwegian HPV reference laboratory at Akershus University Hospital for their molecular expertise and technical support and the Parirenyatwa Hospital VIAC clinic staff for assisting during the recruitment phase. We are also very grateful to Prof. Jane R. Montealegre and all the other reviewers for their constructive critique and suggestions.

Where authors are identified as personnel of the International Agency for Research on Cancer/World Health Organization, the authors alone are responsible for the views expressed in this article and they do not necessarily represent the decisions, policy or views of the International Agency for Research on Cancer /World Health Organization.

## Author Contributions

**Conceptualization:** Racheal S. Dube Mandishora, Babill Stray-Pedersen, Joel Palefsky, Zvavahera M. Chirenje.

**Data curation:** Racheal S. Dube Mandishora, Trine B. Rounge, Ole Herman Ambur, Sonja Lagström.

**Formal analysis:** Racheal S. Dube Mandishora, Sonja Lagström.

**Funding acquisition:** Racheal S. Dube Mandishora, Babill Stray-Pedersen.

**Investigation:** Racheal S. Dube Mandishora.

**Methodology:** Racheal S. Dube Mandishora, Trine B. Rounge, Irene Kraus Christiansen, Sonja Lagström, Babill Stray-Pedersen, Joel Palefsky, Zvavahera M. Chirenje.

**Project administration:** Racheal S. Dube Mandishora.

**Resources:** Racheal S. Dube Mandishora, Zvavahera M. Chirenje.

**Supervision:** Trine B. Rounge, Ole Herman Ambur, Babill Stray-Pedersen, Massimo Tommasino, Joel Palefsky, Zvavahera M. Chirenje.

**Validation:** Megan Fitzpatrick, Irene Kraus Christiansen, Ole Herman Ambur, Massimo Tommasino.

**Visualization:** Racheal S. Dube Mandishora, Megan Fitzpatrick, Massimo Tommasino.

**Writing – original draft:** Racheal S. Dube Mandishora, Trine B. Rounge, Megan Fitzpatrick.

**Writing – review & editing:** Racheal S. Dube Mandishora, Trine B. Rounge, Megan Fitzpatrick, Irene Kraus Christiansen, Ole Herman Ambur, Sonja Lagström, Massimo Tommasino, Zvavahera M. Chirenje.

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
