## [Decision Letter · Decision Letter 0]

10 Aug 2020

PONE-D-20-19224

Self-collected and clinician-collected anal swabs show modest agreement for HPV genotyping

PLOS ONE

Dear Dr. Dube Mandishora,

Thank you for submitting your manuscript to PLOS ONE. After careful consideration, we feel that it has merit but does not fully meet PLOS ONE’s publication criteria as it currently stands. Therefore, we invite you to submit a revised version of the manuscript that addresses the points raised during the review process.

In particular, please ensure that your statistical methods are fully described and appropriately support your study design. Address any inconsistencies noted between your conclusions and the existing body of literature. Finally, be sure to correct any noted spelling or grammatical errors in the manuscript.

We look forward to receiving your revised manuscript.

Kind regards,

Michael Scheurer, Ph.D.

Academic Editor

PLOS ONE

Journal Requirements:

3. In your Methods section, please provide additional information about the participant recruitment method and the demographic details of your participants. Please ensure you have provided sufficient details to replicate the analyses such as:  a) a description of any inclusion/exclusion criteria that were applied to participant recruitment, b) a statement as to whether your sample can be considered representative of a larger population, c) a description of how participants were recruited, and d) descriptions of where participants were recruited and where the research took place.

4. Please provide a sample size and power calculation in the Methods, or discuss the reasons for not performing one before study initiation.

5. To comply with PLOS ONE submission guidelines, in your Methods section, please provide additional information regarding your statistical analyses, including the threshold set for statistical significance for the analyses. For more information on PLOS ONE's expectations for statistical reporting, please see https://journals.plos.org/plosone/s/submission-guidelines.#loc-statistical-reporting.

6. In the Methods section, please provide the specific sequences of the primers used in the next-generation sequencing analysis.

Reviewers' comments:

Reviewer #1: This is an interesting manuscript describing concordance between self- and clinician-collected anal swabs for HPV DNA testing. Given increasing incidence of anal cancer worldwide and new guidelines for anal HPV screening, self-sampling may help screening programs achieve greater coverage. Thus, there is a need for research regarding utility of self-sampling vis a vis clinician sampling. I have the following comments and suggestions:

Abstract

I suggest changing the last sentence to, “Our data suggest that self-collected anal swabs can be used as an alternative to clinician-collected anal swabs for HPV DNA testing in a clinical setting.”

Intro

Paragraph 1 would benefit from mention of burden of disease. I suggest including data on burden of cervical cancer, narrowing in on burden is sub-Saharan Africa.

Paragraph 2 “Although anal cancers are relatively rare, data from USA reports a gradual increase in the incidence of anal cancers especially in HIV-infected individuals, regardless of the anti

retroviral (ART) era [5]. Furthermore, a cumulative incidence of 0.1% by age 75 years was

reported in HIV uninfected individuals as compared to 1.5% in HIV infected individuals.

The increase in anal cancer seems to be broader than among HIV-infected individuals. I suggest revising to: Although anal cancers are relatively rare, data from USA indicates a significant increase in the incidence of anal cancers [REF: https://doi.org/10.1093/jnci/djz219]. The increase is particularly high among HIV-infected individuals in the anti-retroviral era[include data on increasing incidence].

Sidenote: “The increase is particularly high among HIV-infected individuals, regardless of the antiretroviral (ART) era [5].” I’m not sure that it’s “regardless” of the anti-retroviral era, rather than “in” the antiretroviral era. Isn’t the extension of life due to ART driving increasing cancer incidence among HIV+ individuals?

Paragraph 1 and 2 jump around quite a bit. Also, since the paper doesn’t seek to fill the gap in data regarding burden of anal cancer in Zimbabwe, I don’t think it makes sense to set this up as an argument in the Intro. How about ending paragraph 1 with “Women with a history of cervical cancer, especially those with HPV/HIV co-infection, are at increased risk of developing anal cancers [4]. Screening for anal cancer is thus recommended for women with cervical lesions and/or who are HIV infected [9, 10].” And starting paragraph 2 with “Although anal cancers are relatively rare, data from USA indicates a significant increase in the incidence of anal cancers. The increase is particularly high among HIV-infected individuals in the anti-retroviral era. Zimbabwe has an HIV prevalence of almost 13% and cervical cancer contributes a third of all cancers among black Zimbabwean women. In 2016, the Zimbabwe Cancer Registry reported a gradual increase in anal cancers recorded in the past decade. Despite the increased prevalence of risk factors in the population and evidence of increasing incidence, anal cancer screening guidelines are not routinely followed in Zimbabwe.”

Given focus of paper on HPV genotypes and multiple infections, it would be helpful to reader to understand why different genotypes matter. Could this be woven into the intro? This is mentioned in Discussion, but I’d move to Intro

Paragraph 3 jumps between accuracy and acceptability and between anal and cervical swabs. I suggest walking the reader through the arguments to better set up the rationale for the study. Eg. “Utility of self-sample cervical swabs for HPV DNA testing has been demonstrated. Furthermore, self-collection of cervical swabs has shown to be highly acceptable. For anal screening, there is evidence that self-sampling is acceptable and patients tend to be more comfortable and less embarrassed when an anal swab is self-collected. Its clinical utility has been demonstrated for non-HPV studies [sidenote: which tests?] and for cytology. However, here is still scarce data with regards to self-collected sample for HPV DNA testing.”

Methods

Laboratory Methods—DNA extraction. Is this per manufacturers’ instruction? If so, I ‘m not sure that this level of detail is needed.

Results

The results and (discussion) seem to jump back and forth between describing HPV genotypes and presence of HPV DNA. As written, I can’t follow where kappa statistic is for presence of HPV DNA or for specific genotypes.

Discussion

Postulate 3 seems very unlikely.

Paragraphs 7 and 8 (lines 338-353) seem to go beyond the data and focus of the paper. Given this, and that Discussion gets quite lengthy, I’d suggest omitting these paragraphs.

Conclusion: “Cautiously recommend…” should be replaced with “Our data suggest that…” (see comments on abstract).

Reviewer #2: The authors of this generally well-written manuscript were interested in assessing potential anal cancer screening methods that focus on exfoliated cell collection from the anal canal. They compared clinician-collected vs self-collected anal swabs (CCAS vs SCAS) from women attending a cervical cancer screening program in Zimbabwe. The analysis found higher HPV prevalence in SCAS but a higher number of genotypes in CCAS. The method for HPV detection was next generation sequencing and this method leads to a number of questions about the results. At times, the attention to differing options for cut-off read numbers (100 vs 500 vs 1000) obscures the HPV results and calls into question the method of detecting HPV. And if the paper doesn’t have a sound method for detecting HPV, then the HPV results that depend on that detection method, e.g., comparisons of CCAS and SCAS, are called into question. Overall recommendation is to decide which cut-off level is best, defend that decision in the methods with literature, and then report results only for that cut-off level. Until that cut-off level is understood to be clearly the right one, then recommendations for anal cancer screening based on these data may be inappropriate. In addition, since anal cancer screening methods is the reason this study was done, the analysis needs to pay more attention to reporting results of high-risk HPV and results within HIV-positive people.

Abstract:

L53 – focus these results on high-risk types and difference by HIV status.

Methods:

L134 – why were Dacron swabs chosen over other options like flocked swab?

L135 – who did the breaking of the swab into the cryovial? Participant or staff?

L177 – why is 500 better than 100 or 1000? It seems to me that the results can have very different interpretations if, say 100, is chosen instead of 500. There is speculation about false-positive and false-negative results in the discussion, but it seems this should be clearly laid out by the authors in the methods.

L180 – it would be nice to see simple agreement in addition to kappa (agreement beyond chance).

L181 – 183 – this sentence is difficult to understand.

L_ - most assays assess whether or not a sample is adequate for testing. How is this assessed with NGS? Is it possible that a participant could have waived the swab in the air and this swab was judged to be HPV-negative by the authors?

Results:

Table 1 – what kind of contraception for the Yes group? Does the multiple sexual partners variable mean the participant acknowledged having >1 concurrent partner at recruitment?

Supp table 2 I don’t understand the heading: “Number of positive CCAS and SCAS GENOTYPES with cut-off……” Do you mean, “Number of HPV positive CCAS and SCAS specimens with cut-off……”?

Figure 1 A CCAS - HPV16 is typically the most prevalent genotype in other studies as it almost is for CCAS at the 100 cut-off. And it certainly is the most important type for anal cancer. Why not give preference to the most sensitive scenario for detecting HPV16? i.e., choose 100 as the cut-off. The authors state there may be too many false-positives; but the paper doesn’t seem to have a convincing argument that these HPV16 reads are mostly false-positive. Also, since you’re reporting data for three cut-offs, and this is not a methods paper about cut-offs, the message of this figure is that reasonable persons might believe that any one of these 3 cut-offs is closer to the truth.

Fig 2 A and B – the grid lines fall in odd intervals making it tough to assess the frequency for any given column. B – This figure makes no sense to me. It does not communicate the level of concordance in any way that I can understand.

L207 – “type distributions were similar regardless of …cutoffs”….this doesn’t seem to be true since HPV16 – the most important type for anal cancer - has a prevalence of ~13% in CCAS with 100 reads. Prevalence of HPV 16 is otherwise 5 or 6 percent.

L209 – reason for choosing 500 over 100 cutoff is very briefly addressed but reason for choosing 500 over 1000 is not addressed. Again, it seems this should be addressed in the methods.

L221 – overall positivity for each swabbing group is central to this paper and should appear in a table or figure. There are too many data in the results narrative that are not included in a table or figure. In this area I would also expect to see results that are specific to high-risk types since it’s a cancer outcome that the authors care about most.

L227 – why calculate kappa for all genotypes when only a few are responsible for almost all anal cancer?

L252 – It would be nice to see these data in a table stratified by HIV status.

Discussion:

L275 – Other studies have also compared CCAS and SCAS for cytology – Chin-Hong, 2008 and Cranston, 2004

L302 – Add citation for claim that HPV53 is possibly carcinogenic.

L312-313. If the observations are dependent as you suggest (which makes sense), then the kappa statistic is not appropriate since it requires independent observations. Add this to limitations.

L328 – do you mean 40% of women with HPV had multiple infections?

L360 – this range is troubling, because it implies that the McNemar test was almost significant at one of the cut-off thresholds (I presume 100), i.e., at one of the cut-off levels, the p-value was 0.06 for a difference between CCAS and SCAS.

L361 – the authors indicate here that they have adjusted the methods of the study (i.e., chosen a preferred cut-off threshold) in order to maintain statistical significance in their analysis. This is not appropriate. It brings me back to my central concern: that the authors need to justify which cut-off is most appropriate with literature. And if that’s not possible, then maybe a methods paper (that does not make recommendations on anal cancer screening) is most appropriate for these data.

L366. Other than limitation of funds, there is not a clear listing of limitations of this study. One of these might be that kappa assumes the two raters are equally competent which is not the case here.

---

## [Author Response · Author response to Decision Letter 0]

17 Nov 2020

Dear Dr Michael Scheurer

We would like to thank you and the reviewers for the thorough assessment of our work and for the constructive feedback. In the following, we respond to all concerns and comments point-by-point. 

Editor comments

Journal Requirements

Editor point 1: Please ensure that your manuscript meets PLOS ONE's style requirements, including those for file naming. The PLOS ONE style templates can be found at

Reply: We have now formatted and renamed our documents to ensure that they meet the journal’s style. 

Editor point 2: Please include additional information regarding the survey or questionnaire used in the study and ensure that you have provided sufficient details that others could replicate the analyses. For instance, if you developed a questionnaire as part of this study and it is not under a copyright more restrictive than CC-BY, please include a copy, in both the original language and English, as Supporting Information.

Reply: The questionnaire used in our study, written in both English and Shona, is now included as supplementary document 1. We have also indicated the inclusion of the document under section 2.1 L127-128. 

Editor point 3: In your Methods section, please provide additional information about the participant recruitment method and the demographic details of your participants. Please ensure you have provided sufficient details to replicate the analyses such as: a) a description of any inclusion/exclusion criteria that were applied to participant recruitment. b) a statement as to whether your sample can be considered representative of a larger population, c) a description of how participants were recruited, and d) descriptions of where participants were recruited and where the research took place.

Reply: a-d) A line that briefly states that sexually active women aged at least 18 years old were reporting for routine cervical cancer screening were included is now captured in section 2.2 L145-150, whilst L124-129 inform the reader of how and where the study was carried out and to what extent the sample set was representative of a bigger population.

Editor point 4: Please provide a sample size and power calculation in the Methods, or discuss the reasons for not performing one before study initiation. 

Reply: The sample size calculation is now detailed in section 2.1 L130-138. The formular accounted for a power of 0.80.

Editor point 5: To comply with PLOS ONE submission guidelines, in your Methods section, please provide additional information regarding your statistical analyses, including the threshold set for statistical significance for the analyses. For more information on PLOS ONE's expectations for statistical reporting, please see https://journals.plos.org/plosone/s/submission-guidelines.#loc-statistical-reporting. 

Reply: The authors have a working R-script that can be made available upon the readers’ requests directly from the corresponding author. However a summary of the analyses performed on R-Studio is now included on section 2.2.2 L192-213.

Editor point 6: In the Methods section, please provide the specific sequences of the primers used in the next-generation sequencing analysis.

Reply: We have now included specific sequences of the primers as Table 1. L188

Reviewer #1 comments

Reviewer #1: This is an interesting manuscript describing concordance between self- and clinician-collected anal swabs for HPV DNA testing. Given increasing incidence of anal cancer worldwide and new guidelines for anal HPV screening, self-sampling may help screening programs achieve greater coverage. Thus, there is a need for research regarding utility of self-sampling vis a vis clinician sampling. I have the following comments and suggestions:

Abstract

Reviewer #1: I suggest changing the last sentence to, “Our data suggest that self-collected anal swabs can be used as an alternative to clinician-collected anal swabs for HPV DNA testing in a clinical setting.”

Reply: We have now incorporated the suggested wording on L61-63 

Introduction

Reviewer #1: Paragraph 1 would benefit from mention of burden of disease. I suggest including data on burden of cervical cancer, narrowing in on burden is sub-Saharan Africa.

Reply: We have now included facts on the burden of disease in Low-middle income countries and in Zimbabwe. This is captured on L75-78.

Reviewer #1: Paragraph 2 “Although anal cancers are relatively rare, data from USA reports a gradual increase in the incidence of anal cancers especially in HIV-infected individuals, regardless of the antiretroviral (ART) era [5]. Furthermore, a cumulative incidence of 0.1% by age 75 years was reported in HIV uninfected individuals as compared to 1.5% in HIV infected individuals.

The increase in anal cancer seems to be broader than among HIV-infected individuals. I suggest revising to: Although anal cancers are relatively rare, data from USA indicates a significant increase in the incidence of anal cancers [REF: https://doi.org/10.1093/jnci/djz219]. The increase is particularly high among HIV-infected individuals in the anti-retroviral era[include data on increasing incidence].

Side note: “The increase is particularly high among HIV-infected individuals, regardless of the antiretroviral (ART) era [5].” I’m not sure that it’s “regardless” of the anti-retroviral era, rather than “in” the antiretroviral era. Isn’t the extension of life due to ART driving increasing cancer incidence among HIV+ individuals?

Reply: We are grateful for the suggestions and we have now incorporated them on L84-86. We have also cited the paper by Deshmukh et al. Regarding the side note, we used ‘regardless of the ART era’ to indicate that in as much as ART is meant to improve the immunological profiles of an individual, we still have consistent HPV infections leading to cancer. The reviewer points out a different school of thought that describes the extension of life, due to ART, leading to cancer development. Both points are now integrated and captured in L85-88.

Reviewer #1: Paragraph 1 and 2 jump around quite a bit. Also, since the paper doesn’t seek to fill the gap in data regarding burden of anal cancer in Zimbabwe, I don’t think it makes sense to set this up as an argument in the Intro. How about ending paragraph 1 with “Women with a history of cervical cancer, especially those with HPV/HIV co-infection, are at increased risk of developing anal cancers [4]. Screening for anal cancer is thus recommended for women with cervical lesions and/or who are HIV infected [9, 10].” And starting paragraph 2 with “Although anal cancers are relatively rare, data from USA indicates a significant increase in the incidence of anal cancers. The increase is particularly high among HIV-infected individuals in the anti-retroviral era. Zimbabwe has an HIV prevalence of almost 13% and cervical cancer contributes a third of all cancers among black Zimbabwean women. In 2016, the Zimbabwe Cancer Registry reported a gradual increase in anal cancers recorded in the past decade. Despite the increased prevalence of risk factors in the population and evidence of increasing incidence, anal cancer screening guidelines are not routinely followed in Zimbabwe.”

Reply: Paragraph 1 now ends with the sentence on screening for anal cancer and paragraph 2 begins with the incidence of anal cancer in USA. L82-86.

Reviewer #1: Given focus of paper on HPV genotypes and multiple infections, it would be helpful to reader to understand why different genotypes matter. Could this be woven into the intro? This is mentioned in Discussion, but I’d move to Intro

Reply: We agree with the reviewer, for the reader to grasp the importance of reporting HPV genotypes right from the beginning, we have now moved the lines from discussion to the last paragraph of the introduction. L112-119.

Reviewer #1: Paragraph 3 jumps between accuracy and acceptability and between anal and cervical swabs. I suggest walking the reader through the arguments to better set up the rationale for the study. Eg. “Utility of self-sample cervical swabs for HPV DNA testing has been demonstrated. Furthermore, self-collection of cervical swabs has shown to be highly acceptable. For anal screening, there is evidence that self-sampling is acceptable and patients tend to be more comfortable and less embarrassed when an anal swab is self-collected. Its clinical utility has been demonstrated for non-HPV studies [sidenote: which tests?] and for cytology. However, here is still scarce data with regards to self-collected sample for HPV DNA testing.”

Reply: With the reviewer 1’s suggested phrasing, we have now re-written paragraph 3 and improved on the general flow. L95-100. 

Methods

Reviewer #1: Laboratory Methods—DNA extraction. Is this per manufacturers’ instruction? If so, I ‘m not sure that this level of detail is needed.

Reply: Yes, the extraction was as per manufacturer’s instruction. The extra detail on DNA extraction has now been deleted L164-166

Results

Reviewer #1: The results and (discussion) seem to jump back and forth between describing HPV genotypes and presence of HPV DNA. As written, I can’t follow where kappa statistic is for presence of HPV DNA or for specific genotypes.

Reply: To avoid losing the reader and to improve on the flow we have now used consistent terminology, “HPV genotypes” in all sections. The Kappa test is for presence of any HPV genotype.

Discussion

Reviewer #1: Postulate 3 seems very unlikely. 

Reply: We have kept postulate 3 in the text and added the word “unlikely” to guide the readers. L332

Reviewer #1: Paragraphs 7 and 8 (lines 338-353) seem to go beyond the data and focus of the paper. Given this, and that Discussion gets quite lengthy, I’d suggest omitting these paragraphs. 

Reply: Paragraphs 7 describes the distribution of HPV genotypes by HIV status and paragraph 8 briefly describes cervical disease (as detected by Visual-inspection-with-acetic acid). Inasmuch as these two paragraphs mainly discuss secondary data, they are important for the reader to have a full perspective of the sample population. Given that women who have a history of cervical lesions and/or HIV co-infection are at increased risk of anal HPV infection. We have re-written the beginning of paragraph 8, L342, to capture the relevance of the paragraphs. 

Reviewer #1: Conclusion: “Cautiously recommend…” should be replaced with “Our data suggest that…” (see comments on abstract). 

Reply: We agree with the reviewer, therefore the conclusion is now phrased as ‘our data suggest that….’ L61 and L375.

Reviewer #2 comments

Reviewer #2: The authors of this generally well-written manuscript were interested in assessing potential anal cancer screening methods that focus on exfoliated cell collection from the anal canal. They compared clinician-collected vs self-collected anal swabs (CCAS vs SCAS) from women attending a cervical cancer screening program in Zimbabwe. The analysis found higher HPV prevalence in SCAS but a higher number of genotypes in CCAS. The method for HPV detection was next generation sequencing and this method leads to a number of questions about the results. At times, the attention to differing options for cut-off read numbers (100 vs 500 vs 1000) obscures the HPV results and calls into question the method of detecting HPV. And if the paper doesn’t have a sound method for detecting HPV, then the HPV results that depend on that detection method, e.g., comparisons of CCAS and SCAS, are called into question. Overall recommendation is to decide which cut-off level is best, defend that decision in the methods with literature, and then report results only for that cut-off level. Until that cut-off level is understood to be clearly the right one, then recommendations for anal cancer screening based on these data may be inappropriate. In addition, since anal cancer screening methods is the reason this study was done, the analysis needs to pay more attention to reporting results of high-risk HPV and results within HIV-positive people.

Abstract

Reviewer #2: L53 – focus these results on high-risk types and difference by HIV status. 

Reply: We summarise the detection of high-risk HPV genotypes in the discussion section, L302-311 and L348-351. The primary aim and scope of this manuscript was to determine whether the same HPV genotypes were detected in self-collected and clinician-collected swabs, therefore we reported all the genotypes, regardless of whether they are high risk or low-risk. In addition, the most detected genotypes did not include a large number of high-risk genotypes. This is probably because these were mostly ‘disease free’ women reporting for routine screening, a good sample population for the purposes of primarily comparing CCAS to SCAS. We agree that in future studies, where we will focus on HPV genotype distribution other than comparison of collection methods, we can explore in-depth the anal high-risk HPV genotypes and HIV status.

Methods

Reviewer #2: L134 – why were Dacron swabs chosen over other options like flocked swab?

Reply: Our choice was based on literature that reported dacron and flocked swabs having harvested similar amounts of cells. Although in some scenarios flocked swabs are deemed superior, the difference with dacron swabs is minimal. https://www.ncbi.nlm.nih.gov/pmc/articles/PMC5717820/ and https://pubmed.ncbi.nlm.nih.gov/21791907/. In our setting dacron swabs are the most readily available synthetic swabs, therefore given their proven use, we chose them to avoid inconsistency. The likelihood of running out of flocked swabs and having to wait for very long periods for new procurements would have been high. The citations used for justifying the choice of dacron swabs are now included on L145.

Reviewer #2: L150 – who did the breaking of the swab into the cryovial? Participant or staff?

Reply: The collection procedure was explained to all participants by the recruiting nurse. Self-collected swabs were broken into the tube by the participant, immediately after swabbing. Clinician-collected swabs were broken into the tube by the clinician as soon as they completed the swabbing. This is now indicated under section 2.2 L155.

Reviewer #2: L177 – why is 500 better than 100 or 1000? It seems to me that the results can have very different interpretations if, say 100, is chosen instead of 500. There is speculation about false-positive and false-negative results in the discussion, but it seems this should be clearly laid out by the authors in the methods. 

Reply: We agree with the reviewer that presenting more than 1 cut-off can be confusing to the readers, thus we have carefully chosen to report all our findings based on a 500 read pairs cut-off. Our choice is based on the analyses of our read pairs which had a mean of 26232 for CCAS and 19110 for SCAS. Due to the difference in means, instead of normalising we calculated for a slightly higher cut-off. Assuming limit of detection of 5%, our 500 cut-off allows us to filter any false positive results and aligns with clinical recommendations of 300-500 published by Petrackova and colleagues https://pubmed.ncbi.nlm.nih.gov/31552176/. Although, there are a few papers that suggest varied cut-offs, Meisal et al DOI:10.1371/journal.pone.0169074 and Cullen et al doi: 10.1016/j.pvr.2015.05.004 , there is generally no consensus on the minimum required read counts using deep targeted sequencing therefore each laboratory sets its own parameters based on the depth of coverage, the average read pairs and the statistical analyses methods employed. We have now clearly presented our justification (L199-202) and consistently only reported results using a cut-off that’s at least 500 read pairs. Fortunately, one of the co-authors is now preparing a methods paper to suggest the best cut-offs for our laboratory.

Reviewer #2: L180 – it would be nice to see simple agreement in addition to kappa (agreement beyond chance).

Reply: We present both the Cohen’s Kappa and the McNemer because they complement each other. Kappa assumes the two raters are equally competent whilst McNemar allows for unequal raters. We agree that the Kappa assumes the observed agreement to be due to chance. Therefore we included a McNemar test to enhance the agreement analysis and to ensure a more valid interpretation, L210. The McNemar assumes that HPV genotypes detected from self-collected swabs are equal to those detected by clinician-collected swabs. To the best of our knowledge, the complementary effect of these two tests is sufficient to describe the agreement of the collection methods https://pubmed.ncbi.nlm.nih.gov/18188809/ and https://pubmed.ncbi.nlm.nih.gov/10520333/. 

Reviewer #2: L181 – 183 – this sentence is difficult to understand. 

Reply: The sentence has now been rephrased. We have now specifically described how the plots were made using the HPV genotype frequencies. L201-203.

Reviewer #2: L_ - most assays assess whether or not a sample is adequate for testing. How is this assessed with NGS? 

Reply: NGS detects betaglobin as part of a control step to check if the DNA is adequate for testing. However, our samples were treated differently because they had already undergone betaglobin testing as an upstream procedure for a different assay (dot-blot-hybridisation) in Zimbabwe, for a different research scope. This was not repeated before NGS. We have now indicated this in the methods section. L166-168 

Reviewer #2: Is it possible that a participant could have waived the swab in the air and this swab was judged to be HPV-negative by the authors? 

Reply: This is highly unlikely, because the participants had the procedure explained to them by the nurse, who also closely monitored any women who requested supervision. As soon as the swab was collected it was transferred into the cryotube.

Results

Reviewer #2: Table 1 – what kind of contraception for the Yes group? 

Reply: This question referred to any type of contraception. A follow-up question also captured the specific contraception; however, this detailed information is not relevant for the scope of this manuscript. Please see question 14 of the questionnaire, attached as supplementary document 1.

Reviewer #2: Does the multiple sexual partners variable mean the participant acknowledged having >1 concurrent partner at recruitment? 

Reply: Yes, this is correct. The participants self-reported having more than 1 sexual partners at the time of recruitment. Please see the questionnaire attached as supplementary material on question numbers 9 to 11.

Reviewer #2: Supp table 2 I don’t understand the heading: “Number of positive CCAS and SCAS GENOTYPES with cut-off……” Do you mean, “Number of HPV positive CCAS and SCAS specimens with cut-off……”? 

Reply: Thank you for this note, this was a typographic error. The caption for supplementary table 2 (which is now table 3) has now been corrected to “specimens” instead of “genotypes”. L226

Reviewer #2: Figure 1 A CCAS - HPV16 is typically the most prevalent genotype in other studies as it almost is for CCAS at the 100 cut-off. And it certainly is the most important type for anal cancer. Why not give preference to the most sensitive scenario for detecting HPV16? i.e., choose 100 as the cut-off. The authors state there may be too many false-positives; but the paper doesn’t seem to have a convincing argument that these HPV16 reads are mostly false-positive. Also, since you’re reporting data for three cut-offs, and this is not a methods paper about cut-offs, the message of this figure is that reasonable persons might believe that any one of these 3 cut-offs is closer to the truth.

Reply: We have considered your suggestions and settled for a cut-off of 500. This is based on the explanation given above, for your question on L177. Concerning HPV 16 prevalence, indeed there seems to be a reduction in the cases but it was not statistically significant. Furthermore, with NGS we expect the sensitivity to be standard for all genotypes. We are also of the view that presenting an optimum cut-off will be better regardless of the loss of seemingly ‘positive’ samples because if they do not meet the optimum cut-off, we consider them to be negative. We also agree that presenting more than one cut-off is not suited for the scope of this paper. These will now be presented in a different methods manuscript that some of the co-authors are working on. To avoid misleading the readers, we have removed Figure 1. This will be best suited for a methods manuscript.

Reviewer #2: Fig 2 A and B – the grid lines fall in odd intervals making it tough to assess the frequency for any given column. B – This figure makes no sense to me. It does not communicate the level of concordance in any way that I can understand.

Reply: The authors have taken a closer look at figure 2 (which is now renamed to Figure 1) and we agree that the way the frequencies were presented may have been confusing. We have now relabelled the y-axis to present frequencies as whole numbers as opposed to the 0.5 intervals. The main message conveyed by the figure is the difference of frequencies of the HPV genotypes detected by each method (A), whilst (B) further gives a pictorial illustration of instances when an HPV genotype was detected by either of the collection methods or by both. 

Reviewer #2: L207 – “type distributions were similar regardless of …cutoffs”….this doesn’t seem to be true since HPV16 – the most important type for anal cancer - has a prevalence of ~13% in CCAS with 100 reads. Prevalence of HPV 16 is otherwise 5 or 6 percent. L209 – reason for choosing 500 over 100 cutoff is very briefly addressed but reason for choosing 500 over 1000 is not addressed. Again, it seems this should be addressed in the methods.

Reply: Similar to the response to Figure 1 above. We have now justified the use of 500 as a cut-off, therefore we have now discarded the need to compare HPV genotypes based on cut-offs, as we agree that this is not a methods paper.

Reviewer #2: L221 – overall positivity for each swabbing group is central to this paper and should appear in a table or figure. There are too many data in the results narrative that are not included in a table or figure. In this area I would also expect to see results that are specific to high-risk types since it’s a cancer outcome that the authors care about most. 

Reply: Regarding the overall positivity for each swabbing group, we have now presented these in table 3 L229, previously labelled as supplementary table 1 and 2. We agree with the reviewer, cancer outcome is a critical end point which we wish to prevent by providing data that can offer more screening options. In the context of cancer development, high-risk genotypes have the most clinical relevance. However, the primary aim and scope of this manuscript was to determine whether the same HPV genotypes were detected in self-collected and clinician-collected swabs, therefore we reported all the genotypes including those with less clinical relevance. Having determined and described the agreement using a broad array of HPV genotypes, we envision that when applications are being made in a clinical setting, the clinician will receive a report specifying the clinically relevant HPV genotypes. 

Reviewer #2: L227 – why calculate kappa for all genotypes when only a few are responsible for almost all anal cancer?

Reply: The reviewer has rightfully highlighted that high-risk HPV genotypes have the most clinical relevance. We used Illumina next-generation-sequencing which detects a broad array of HPV genotypes, thus our overall HPV anal tests should report good accuracy for all HPV genotypes. The primary aim and scope of this manuscript was to determine whether the same HPV genotypes were detected in self-collected and clinician-collected swabs, therefore we reported all the genotypes including those with less clinical relevance. 

Reviewer #2: L252 – It would be nice to see these data in a table stratified by HIV status.

Reply: The scope of this manuscript is to compare two methods and we do not expect the HIV status of the women to affect the accuracy of the test, therefore we do not present the HPV genotype agreement of CCAS and SCAS by HIV status. However, we give a summary description of the HIV status of the sample population based on the gold-standard, which is the clinician-collected in this case. This gives the reader a good indication of the HPV/HIV co-infection in this group of women. Sections 3.3 L260-270.

Discussion

Reviewer #2: L275 – Other studies have also compared CCAS and SCAS for cytology – Chin-Hong, 2008 and Cranston, 2004

Reply: Thank you for highlighting these important studies to us. The line has now been rephrased and the citations are now updated L98 and L282-283.

Reviewer #2: L302 – Add citation for claim that HPV53 is possibly carcinogenic.

Reply: A citation (Meyer et al 2001) has now been added and the phrase is also slightly modified to “probable high-risk”. L309-310.

Reviewer #2: L312-313. If the observations are dependent as you suggest (which makes sense), then the kappa statistic is not appropriate since it requires independent observations. Add this to limitations. 

Reply: We have now highlighted this limitation on L363-370.

Reviewer #2: L328 – do you mean 40% of women with HPV had multiple infections?

Reply: Yes, 40% of women with HPV had multiple infections. We have added the phrase “of HPV infected” to clearly capture this L335.

Reviewer #2: L360 – this range is troubling, because it implies that the McNemar test was almost significant at one of the cut-off thresholds (I presume 100), i.e., at one of the cut-off levels, the p-value was 0.06 for a difference between CCAS and SCAS. 

Reply: Thank you for this note. We have chosen our best cut-off to be 500, as described in the earlier responses in this document. We have now removed the comparisons that may confuse the reader. We agree that at one of the cut-offs the p-value of 0.06 was boarder line significant, therefore it should be confirmed using a larger sample size in other future studies. 

Reviewer #2: L361 – the authors indicate here that they have adjusted the methods of the study (i.e., chosen a preferred cut-off threshold) in order to maintain statistical significance in their analysis. This is not appropriate. It brings me back to my central concern: that the authors need to justify which cut-off is most appropriate with literature. And if that’s not possible, then maybe a methods paper (that does not make recommendations on anal cancer screening) is most appropriate for these data.

Reply: We agree that our inferences may have not been well spelt out. The comparison of cut-offs was not appropriate for this paper. We will publish such comparisons in a different paper. To avoid any misinterpretations by the reader we have now presented all our data based on the optimum cut-off of 500 as justified in the response to similar comments above. 

Reviewer #2: L366. Other than limitation of funds, there is not a clear listing of limitations of this study. One of these might be that kappa assumes the two raters are equally competent which is not the case here. 

Reply: We have now collated our limitations and presented them on L360-366 and L363-370. We present the Kappa and McNemer because they complement each other. Kappa assumes the two raters are equally competent whilst McNemar allows for unequal raters.

---

## [Decision Letter · Decision Letter 1]

31 Dec 2020

PONE-D-20-19224R1

Self-collected and clinician-collected anal swabs show modest agreement for HPV genotyping

PLOS ONE

Dear Dr. Dube Mandishora,

Thank you for submitting your manuscript to PLOS ONE. After careful consideration, we feel that it has merit but does not fully meet PLOS ONE’s publication criteria as it currently stands. Therefore, we invite you to submit a revised version of the manuscript that addresses the points raised during the review process.

We look forward to receiving your revised manuscript.

Kind regards,

Michael Scheurer, Ph.D.

Academic Editor

PLOS ONE

Additional Editor Comments (if provided):

In addressing the remaining reviewers' comments below, please also ensure that you have cited the correct references to support the use of the statistical tests that you used.

Please confirm that your manuscript meets the PLOS One data submission/availability policy. In particular, the policy states that "data points used to create, for example, means and medians, need to be available."

Reviewers' comments:

Reviewer's Responses to Questions

**Comments to the Author**

1. If the authors have adequately addressed your comments raised in a previous round of review and you feel that this manuscript is now acceptable for publication, you may indicate that here to bypass the “Comments to the Author” section, enter your conflict of interest statement in the “Confidential to Editor” section, and submit your "Accept" recommendation.

Reviewer #1: (No Response)

Reviewer #2: (No Response)

2. Is the manuscript technically sound, and do the data support the conclusions?

Reviewer #1: Partly

Reviewer #2: Yes

3. Has the statistical analysis been performed appropriately and rigorously? 

Reviewer #1: No

Reviewer #2: Yes

4. Have the authors made all data underlying the findings in their manuscript fully available?

Reviewer #1: Yes

Reviewer #2: (No Response)

5. Is the manuscript presented in an intelligible fashion and written in standard English?

Reviewer #1: Yes

Reviewer #2: Yes

6. Review Comments to the Author

Reviewer #1: The authors have satisfactorily addressed all of my previous comments. However, I have some concerns that I had overlooked regarding the statistical analysis, specifically the use of McNamar’s test as a test of agreement (see below). The authors conclude that there is moderate agreement (based on kappa=0.55), but state that this was not statistically significant based on McNamar’s test. However, McNamar’s is a test of proportions, rather than agreement. Thus, the p-value used to support the conclusion is only able to conclude that the proportion of a particular genotype was not significantly different between CCAS and SCAS, rather than indicating agreement. My other comments relate to minor issues that should be easy to address.

Abstract: There’s a small typo in conclusions: “with no statistically-significant difference.”

Intro:

I appreciate the authors’ clarification of the increasing incidence of anal cancer among HIV-infected individuals in the era of ART: “The increase is particularly high among HIV-infected

individuals regardless of a repaired immune system due to ART, and conversely, extension

to life due to ART tends to give time for cancer to develop [9].” Just to clarify, is the null association at the individual level (HIV+ individuals on ART are as likely than those not on ART to develop anal cancer) or at the ecologic level (despite advances and coverage in ART, anal cancer incidence continues to rise among HIV+)? If the latter, I think it may be better to say something like, “The increase is particularly high among HIV-infected individual despite the availability and widespread (?)/ increasing (?) use of ART, which improves individuals’ immunogenic profile. Conversely, ART extends life among HIV infected individuals, giving time for cancer to develop.”

Also, please define ART on first use

101-103: The last sentence of this paragraph is repeated from above: “Furthermore,

patients tend to be more comfortable and less embarrassed when an anal swab is self

collected [21].

112-13: please define SCAS and CCAS on first use

Methods:

144-148 The information in this paragraph seems redundant with the paragraph below. I suggest omitting this paragraph and integrating any new information provided into the paragraph below.

Agreement was tested using kappa and McNamar’s test. However, McNamar’s test compares overall proportions, not agreement (see: (see: Ranganathan P, Pramesh CS, Aggarwal R. Common pitfalls in statistical analysis: Measures of agreement. Perspect Clin Res. 2017 Oct-Dec;8(4):187-191.). I’m not confident that this test can be used to support the authors’ conclusions.

Results:

Table 3 says CCAS: 124 HPV + samples, SCAS:114 HPV + samples. But paragraph below says 67 and 75, respectively. What is Table 3, last column referring to?

249-250: This sentence is confusing and I’m not sure of relevance: In addition, 13/29 of these women had multiple infections in both SCAS and CCAS regardless of HPV genotype and number of infections.

Figure 3. I’m confused as to why there’s a single p-value for the McNamar’s test. McNamar’s is used for paired nominal data in a 2x2 contingency table. So, for each genotype CCAS vs. SCAS and genotype present versus genotype absent. It seems like there should be a test statistic and associated p-value for each genotype-comparison, rather than a global McNamar’t test statistic and p-value.

Discussion:

327-334: The same has been observed for self-collected versus clinician-collected cervical swabs [See Arbyn et al 2018-- https://pubmed.ncbi.nlm.nih.gov/30518635/], attributed to vaginal sampling with self-collection versus cervical sampling with CC.

The author’s conclude that there is moderate agreement (based on kappa=0.55), but state that this was not statistically significant, based on McNamar’s test. However, McNamar’s isn’t a test of agreement. Thus, I’m not certain that the data support this conclusion.

Reviewer #2: Thanks for addressing my comments. I appreciate the authors complete and thoughtful replies. I would only point out that the following critique does seem to have been addressed in the revised manuscript.

L312-313. If the observations are dependent as you suggest (which makes sense), then the kappa statistic is not appropriate since it requires independent observations. Add this to limitations.

Reply: We have now highlighted this limitation on L363-370.

7. PLOS authors have the option to publish the peer review history of their article (what does this mean?). If published, this will include your full peer review and any attached files.

Reviewer #1: No

Reviewer #2: No

---

## [Author Response · Author response to Decision Letter 1]

14 Feb 2021

Dear Dr Michael Scheurer

We would like to thank you and the reviewers for the additional comments on our work and for the constructive feedback. In the following, we respond to all concerns and comments point-by-point. 

Additional Editor Comments

Comment 1: In addressing the remaining reviewers' comments below, please also ensure that you have cited the correct references to support the use of the statistical tests that you used.

Response: Additional references have now been included, such as Ranganathan P, Pramesh CS, Aggarwal R. Common pitfalls in statistical analysis: Measures of agreement. Perspect Clin Res. 2017 Oct-Dec;8(4):187-191.

Comment 2: Please confirm that your manuscript meets the PLOS One data submission/availability policy. In particular, the policy states that "data points used to create, for example, means and medians, need to be available."

Response: The authors have a working R-script that can be made available upon the readers’ requests directly from the corresponding author.

Reviewer #1 Comments to the Author

Comment 1: The authors have satisfactorily addressed all of my previous comments. However, I have some concerns that I had overlooked regarding the statistical analysis, specifically the use of McNemar’s test as a test of agreement (see below). The authors conclude that there is moderate agreement (based on kappa=0.55), but state that this was not statistically significant based on McNemar’s test. However, McNemar’s is a test of proportions, rather than agreement. Thus, the p-value used to support the conclusion is only able to conclude that the proportion of a particular genotype was not significantly different between CCAS and SCAS, rather than indicating agreement. My other comments relate to minor issues that should be easy to address.

Response: Thank you for highlighting the interpretation of these statistical analyses. To avoid confusion for the readers, the authors have now separated the two statistical analyses to represent (1) measure of agreement (L54-55) and (2) test of proportions (L55). This has also been applied throughout the manuscript in the methods (L223-225), results (L253-255) and discussion (L314-318). The tests are not used as one but rather as two separate complimentary tests.

Comment 2: Abstract: There’s a small typo in conclusions: “with no statistically-significant difference.”

Response: 

We have now deleted the “none” and used “no” on L58.

Introduction

Comment 3: I appreciate the authors’ clarification of the increasing incidence of anal cancer among HIV-infected individuals in the era of ART: “The increase is particularly high among HIV-infected individuals regardless of a repaired immune system due to ART, and conversely, extension to life due to ART tends to give time for cancer to develop [9].” Just to clarify, is the null association at the individual level (HIV+ individuals on ART are as likely than those not on ART to develop anal cancer) or at the ecologic level (despite advances and coverage in ART, anal cancer incidence continues to rise among HIV+)? If the latter, I think it may be better to say something like, “The increase is particularly high among HIV-infected individual despite the availability and widespread (?)/ increasing (?) use of ART, which improves individuals’ immunogenic profile. Conversely, ART extends life among HIV infected individuals, giving time for cancer to develop.” Also, please define ART on first use 101-103: The last sentence of this paragraph is repeated from above: “Furthermore, patients tend to be more comfortable and less embarrassed when an anal swab is self-collected [21].

Response: Yes, the intention was to explain the latter scenario. The null association is at an ecological level. We have therefore edited the paragraph to include the reviewer’s useful suggestions, L86-88. The acronym ART has now been introduced in L87. The repeated sentence has now been deleted from line 108.

Comment 4: 112-13: please define SCAS and CCAS on first use

Response: The acronyms SCAS and CCAS are now introduced on L121, in addition to the definition that was provided in the abstract L40.

Methods

Comment 5: 144-148 The information in this paragraph seems redundant with the paragraph below. I suggest omitting this paragraph and integrating any new information provided into the paragraph below.

Agreement was tested using kappa and McNamar’s test. However, McNamar’s test compares overall proportions, not agreement (see: (see: Ranganathan P, Pramesh CS, Aggarwal R. Common pitfalls in statistical analysis: Measures of agreement. Perspect Clin Res. 2017 Oct-Dec;8(4):187-191.). I’m not confident that this test can be used to support the authors’ conclusions.

Response: We agree with the reviewer, stating that agreement was tested using kappa and McNemar can be misleading. To remove this confusion, we have now separated the two statistical analyses to represent (1) measure of agreement (L54-55) and (2) test of proportions (L55). This is also explained in the responses above. The authors believe that both statistical tests are useful. We are also conscious of the fact that there is no perfect test to evaluate agreement and it’s a classical source of debate. Both tests are commonly used together to complement each other https://pubmed.ncbi.nlm.nih.gov/10520333/. McNemar is indeed a test of proportion, but the proportion of discordant pairs (ie +/- vs -/+), which gives useful information when evaluating (dis)agreement based on proportions. Indeed, as mentioned by Ranganathan et al, a non-significant McNemar test is not enough to say the agreement is good. It’s however still useful, if interpreted correctly. We have now further clarified our interpretations and limitations in L399-405 of the Discussion section.

Results

Comment 6: Table 3 says CCAS: 124 HPV + samples, SCAS:114 HPV + samples. But paragraph below says 67 and 75, respectively. What is Table 3, last column referring to?

Response: We apologise for the misleading column. The 124 and 114 refer to every HPV positivity regardless of multiple infections. In other words, for CCAS, the total read pairs amounted to 124 HPV genotypes in 67 women. This is because some of the 67 women had multiple HPV genotypes, such that there were 32 different HPV genotypes repeatedly detected in 67 women. However, to avoid confusing the reader, the column is now titled “number of HPV positive women”.

Comment 7: 249-250: This sentence is confusing and I’m not sure of relevance: In addition, 13/29 of these women had multiple infections in both SCAS and CCAS regardless of HPV genotype and number of infections.

Response: We have now removed the statement. 

Comment 8: Figure 3. I’m confused as to why there’s a single p-value for the McNamar’s test. McNamar’s is used for paired nominal data in a 2x2 contingency table. So, for each genotype CCAS vs. SCAS and genotype present versus genotype absent. It seems like there should be a test statistic and associated p-value for each genotype-comparison, rather than a global McNamar’t test statistic and p-value.

Response: The reviewer raises an important question which fortunately we also touched on in the first rebuttal. In our data, each genotype was considered independent because we were most interested in knowing how much HPV was detected in either methods or in comparison. Given the broad array of HPV genotypes that the Illumina sequencer could detect, it was complicated to break down our analyses into each and every HPV genotype. Discordant results were defined as those that were HPV positive on one specimen type while negative on the other. However, we have highlighted this under limitations L294-299 and L371-384. Furthermore, NGS is a highly sensitive assay (DOI:10.1038/s41598-018-36669-6 ) and we do not expect a difference in the capacity for detecting HPV genotypes assuming there are infected cells on the swabs, rather, our research question is whether each collection method harvests enough cells for HPV genotypes to be detected, when comparing two detection methods.

Discussion

Comment 9: 327-334: The same has been observed for self-collected versus clinician-collected cervical swabs [See Arbyn et al 2018-- https://pubmed.ncbi.nlm.nih.gov/30518635/], attributed to vaginal sampling with self-collection versus cervical sampling with CC.

Response: Thank you for this important reference. We have included it in our discussion, L340-342

Comment 10: The authors conclude that there is moderate agreement (based on kappa=0.55), but state that this was not statistically significant, based on McNamar’s test. However, McNamar’s isn’t a test of agreement. Thus, I’m not certain that the data support this conclusion.

Response: We agree with the reviewer. Yes, the McNemar is best referred to as a method to ‘evaluate agreement’, as opposed to a ‘test of agreement’. We have now carefully rephrased the abstract and conclusion to fully portray what both statistical tests do and mean. This has now been applied throughout the manuscript in the methods (L210-212), results (L238-240) and discussion (L314-318). We have also toned down our conclusion to ensure that the readers fully understand what our results mean L287-293.

Reviewer #2 Comments to the Author

Comment 1: Thanks for addressing my comments. I appreciate the authors complete and thoughtful replies. I would only point out that the following critique does seem to have been addressed in the revised manuscript.

L312-313. If the observations are dependent as you suggest (which makes sense), then the kappa statistic is not appropriate since it requires independent observations. Add this to limitations.

Reply: We have now highlighted this limitation on L363-370.

Response: Thank you for the comment, but unfortunately there is probably a mix up on the line numbering and we are not sure if we have fully understood your comment. Nonetheless, as mentioned above, the authors believe that both statistical tests are useful. We are also conscious of the fact that there is no perfect test to evaluate agreement. Both tests are commonly used together to complement each other. However, we have spelt out the limitations in L294-299 and L371-384. If the reviewer is talking about the different hypotheses for higher detection with SC, (because L302 corresponds to L232 on the original pdf and L328 to L259), then the fact that the second swab is influenced by the first one, is just a hypothesis, the observations (SC and CC) can be considered independent. We have reiterated our speculative hypotheses for these differences in L322-323.

---

## [Decision Letter · Decision Letter 2]

19 Mar 2021

PONE-D-20-19224R2

Self-collected and clinician-collected anal swabs show modest agreement for HPV genotyping

PLOS ONE

Dear Dr. Dube Mandishora,

Thank you for submitting your manuscript to PLOS ONE. After careful consideration, we feel that it has merit but does not fully meet PLOS ONE’s publication criteria as it currently stands. Therefore, we invite you to submit a revised version of the manuscript that addresses the points raised during the review process.

Please address the reviewer's suggestions for edits to clarify information presented to the reader in the methods and the discussion sections.

We look forward to receiving your revised manuscript.

Kind regards,

Michael Scheurer, Ph.D.

Academic Editor

PLOS ONE

Journal Requirements:

Reviewers' comments:

Reviewer's Responses to Questions

**Comments to the Author**

1. If the authors have adequately addressed your comments raised in a previous round of review and you feel that this manuscript is now acceptable for publication, you may indicate that here to bypass the “Comments to the Author” section, enter your conflict of interest statement in the “Confidential to Editor” section, and submit your "Accept" recommendation.

Reviewer #1: (No Response)

2. Is the manuscript technically sound, and do the data support the conclusions?

Reviewer #1: Yes

3. Has the statistical analysis been performed appropriately and rigorously? 

Reviewer #1: Yes

4. Have the authors made all data underlying the findings in their manuscript fully available?

Reviewer #1: Yes

5. Is the manuscript presented in an intelligible fashion and written in standard English?

Reviewer #1: Yes

6. Review Comments to the Author

Reviewer #1: The authors have fully addressed all of my comments and suggestions. I have a few minor suggestions related to how the kappa and McNemar’s are presented and a few minor edits in the Discussion.

Abstract (and throughout) Thank you for clarifying that kappa and McNamar’s were two separate tests used to evaluate concordance. For the sake of clarity, could the following edits be made to the abstract:

Methods: Currently: Level of agreement was calculated using the kappa and McNemar tests. Suggested: Level of agreement of HPV genotypes between CCAS and SCAS was calculated using the kappa statistic. McNemar tests were used to evaluate agreement in the proportion of genotypes detected by either method.

The same info is available elsewhere in the abstract, but it would improve clarity to include it in the Methods so that readers can better understand the Results.

Methods: Similarly, I’d clarify how the statistics were used in the last paragraph of the Methods. Currently: To evaluate agreement of the two collection methods, the McNemar non-parametric test was performed. A kappa test for agreement was also performed to complement the McNemar [38–40]. Suggested: A kappa test was used to evaluate agreement of HPV genotypes across the two collection methods. The McNemar’s non-parametric test was performed to evaluated agreement in the proportion of genotypes detected by either method.

Discussion, line 294-300: Current: Based on the moderate level of agreement of HPV genotypes between the two methods, signified by 0.55 in kappa value (k), our data suggest that researchers can use either methods for collection. Furthermore, to evaluate agreement of the methods, McNemar gave a Chi-square value of 0.75 (p=0.39), which showed that the difference in detection rates on the two methods was not statistically significant. Suggested: Based on the moderate level of agreement of HPV genotypes between the two methods, indicated by 0.55 in kappa value (k) and a non-significant test of proportions, our data suggest that researchers can use either methods for collection.

Line 351-353 (minor edit): Arbyn et al had similar observations, reporting HPV detection rates 2.28 times higher in self-collected versus clinician-collected vaginal swabs [45].

Line 384-389, Current: Indeed, as mentioned by Ranganathan et al, a non-significant McNemar test is not always enough to say the agreement is good. It’s however still useful, if interpreted correctly [51]. In our case, the McNemar is indeed a test of proportion, but the proportion of discordant pairs (ie +/- vs -/+), which gave useful information for evaluating (dis)agreement of HPV genotype detection. Suggested: Indeed, as mentioned by Ranganathan et al, a non-significant McNemar test alone is not sufficient to indicate good agreement. However, it is still a useful statistic, if interpreted correctly [51]. In our case, the McNemar test indicated that the proportion of discordant pairs (ie +/- vs -/+) was not statistically significant, providing useful information for evaluating (dis)agreement of HPV genotype detection.

7. PLOS authors have the option to publish the peer review history of their article (what does this mean?). If published, this will include your full peer review and any attached files.

Reviewer #1: **Yes: **Jane R Montealegre

---

## [Author Response · Author response to Decision Letter 2]

31 Mar 2021

Dear Dr Michael Scheurer

We would like to thank you and the reviewers for the additional comments on our work and for the constructive feedback. In the following, we respond to all concerns and comments point-by-point. 

Review # 1 Comments to the Author

Comment 1: The authors have fully addressed all of my comments and suggestions. I have a few minor suggestions related to how the kappa and McNemar’s are presented and a few minor edits in the Discussion.

Abstract (and throughout) Thank you for clarifying that kappa and McNamar’s were two separate tests used to evaluate concordance. For the sake of clarity, could the following edits be made to the abstract:

Methods: Currently: Level of agreement was calculated using the kappa and McNemar tests. Suggested: Level of agreement of HPV genotypes between CCAS and SCAS was calculated using the kappa statistic. McNemar tests were used to evaluate agreement in the proportion of genotypes detected by either method.

The same info is available elsewhere in the abstract, but it would improve clarity to include it in the Methods so that readers can better understand the Results.

Response: We have now included the suggested phrasing on L48-51.

Comment 2: Methods: Similarly, I’d clarify how the statistics were used in the last paragraph of the Methods. Currently: To evaluate agreement of the two collection methods, the McNemar non-parametric test was performed. A kappa test for agreement was also performed to complement the McNemar [38–40]. Suggested: A kappa test was used to evaluate agreement of HPV genotypes across the two collection methods. The McNemar’s non-parametric test was performed to evaluated agreement in the proportion of genotypes detected by either method.

Response: We have now included the suggested phrasing on L213-216.

Comment 3: Discussion, line 294-300: Current: Based on the moderate level of agreement of HPV genotypes between the two methods, signified by 0.55 in kappa value (k), our data suggest that researchers can use either methods for collection. Furthermore, to evaluate agreement of the methods, McNemar gave a Chi-square value of 0.75 (p=0.39), which showed that the difference in detection rates on the two methods was not statistically significant. Suggested: Based on the moderate level of agreement of HPV genotypes between the two methods, indicated by 0.55 in kappa value (k) and a non-significant test of proportions, our data suggest that researchers can use either methods for collection.

Response: We have now included the suggested phrasing on L294-297.

Comment 4: Line 351-353 (minor edit): Arbyn et al had similar observations, reporting HPV detection rates 2.28 times higher in self-collected versus clinician-collected vaginal swabs [45].

Response: We have now added the word “versus” to the sentence, L343-345. 

Comment 5: Line 384-389, Current: Indeed, as mentioned by Ranganathan et al, a non-significant McNemar test is not always enough to say the agreement is good. It’s however still useful, if interpreted correctly [51]. In our case, the McNemar is indeed a test of proportion, but the proportion of discordant pairs (ie +/- vs -/+), which gave useful information for evaluating (dis)agreement of HPV genotype detection. Suggested: Indeed, as mentioned by Ranganathan et al, a non-significant McNemar test alone is not sufficient to indicate good agreement. However, it is still a useful statistic, if interpreted correctly [51]. In our case, the McNemar test indicated that the proportion of discordant pairs (ie +/- vs -/+) was not statistically significant, providing useful information for evaluating (dis)agreement of HPV genotype detection.

Response: We have now rephrased L388-393, to in cooperate the suggestion. 

Additional edits by authors: Additional information on author contributions has now been included on L428-431, to indicate the unfortunate loss of one of our co-authors BSP, who passed away before the submission of the final version. We have also taken the opportunity to thank the reviewers for their contributions in remoulding this manuscript, L408 specifically mentions Prof. Jane Montealegre and those who also remained anonymous.

Do you want your identity to be public for this peer review? For information about this choice, including consent withdrawal, please see our Privacy Policy.

Reviewer #1: Yes: Jane R Montealegre

---

## [Editor Report · Decision Letter 3]

7 Apr 2021

Self-collected and clinician-collected anal swabs show modest agreement for HPV genotyping

PONE-D-20-19224R3

Dear Dr. Dube Mandishora,

We’re pleased to inform you that your manuscript has been judged scientifically suitable for publication and will be formally accepted for publication once it meets all outstanding technical requirements.

Kind regards,

Michael Scheurer, Ph.D.

Academic Editor

PLOS ONE
---

## [Editor Report · Acceptance letter]

12 Apr 2021

PONE-D-20-19224R3 

 Self-collected and clinician-collected anal swabs show modest agreement for HPV genotyping 

Dear Dr. Dube Mandishora:

I'm pleased to inform you that your manuscript has been deemed suitable for publication in PLOS ONE. Congratulations! Your manuscript is now with our production department. 

Kind regards, 

on behalf of

Dr. Michael Scheurer 

Academic Editor

PLOS ONE